# Meta-Analysis Reveals Both the Promises and the Challenges of Clinical Metabolomics

**DOI:** 10.3390/cancers14163992

**Published:** 2022-08-18

**Authors:** Heidi E. Roth, Robert Powers

**Affiliations:** 1Department of Chemistry, University of Nebraska-Lincoln, Lincoln, NE 68588-0304, USA; 2Nebraska Center for Integrated Biomolecular Communication, University of Nebraska-Lincoln, Lincoln, NE 68588-0304, USA

**Keywords:** meta-analysis, pancreatic cancer biomarkers, clinical metabolomics, NMR, mass spectrometry

## Abstract

**Simple Summary:**

A highly desirable approach to diagnose a human disease quickly and easily is to identify a chemical, protein, or antibody in a biofluid like blood or urine that is uniquely associated with the disease state and can serve as a diagnostic biomarker. Metabolomics is the study of the set of metabolites or small molecular-weight compounds found in cells, tissues, organs, and organisms. Thus, metabolomics is being widely applied to a variety of human diseases including various cancers to identify potential metabolite biomarkers for disease diagnosis and personalized medicine. Pancreatic ductal adenocarcinoma (PDAC) is the third-leading cause of cancer-related death and has the lowest five-year survival rate primarily due to the lack of an early diagnosis. A total of 24 metabolomics studies have been reported in the scientific literature that aimed to identify diagnostic biomarkers for PDAC. We analyzed the outcomes from these 24 studies in detail and observed a high level of inconsistencies in the identified metabolites that we attributed to a variety of experimental factors. Despite this negative outcome, we did identify a set of 10 metabolites that were consistently detected by several clinical studies and have the potential to serve as PDAC biomarkers and warrant further investigation.

**Abstract:**

Clinical metabolomics is a rapidly expanding field focused on identifying molecular biomarkers to aid in the efficient diagnosis and treatment of human diseases. Variations in study design, metabolomics methodologies, and investigator protocols raise serious concerns about the accuracy and reproducibility of these potential biomarkers. The explosive growth of the field has led to the recent availability of numerous replicate clinical studies, which permits an evaluation of the consistency of biomarkers identified across multiple metabolomics projects. Pancreatic ductal adenocarcinoma (PDAC) is the third-leading cause of cancer-related death and has the lowest five-year survival rate primarily due to the lack of an early diagnosis and the limited treatment options. Accordingly, PDAC has been a popular target of clinical metabolomics studies. We compiled 24 PDAC metabolomics studies from the scientific literature for a detailed meta-analysis. A consistent identification across these multiple studies allowed for the validation of potential clinical biomarkers of PDAC while also highlighting variations in study protocols that may explain poor reproducibility. Our meta-analysis identified 10 metabolites that may serve as PDAC biomarkers and warrant further investigation. However, 87% of the 655 metabolites identified as potential biomarkers were identified in single studies. Differences in cohort size and demographics, *p*-value choice, fold-change significance, sample type, handling and storage, data collection, and analysis were all factors that likely contributed to this apparently large false positive rate. Our meta-analysis demonstrated the need for consistent experimental design and normalized practices to accurately leverage clinical metabolomics data for reliable and reproducible biomarker discovery.

## 1. Introduction

Molecular biomarkers significantly improve the efficiency and quality of healthcare [1]. They are routinely used for disease detection and prevention, to assess an individual’s risk to acquire or develop a disease, and for monitoring disease progression and a response to treatment [1]. Molecular biomarkers have been used for decades and consist of proteins, genes, and molecules that also includes metabolites [2]. Some classical metabolic biomarkers correspond to glucose (for diabetes) [3], creatinine (for metabolic disorders) [4], and cholesterol and triglyceride (for cardiovascular diseases) [5]. The utility of these and other biomarkers is predicated on the principle that the metabolite is uniquely and reproducibly associated with the disease state. An inherent challenge in biomarker discovery is achieving a true statistically significant correlation between the metabolite and the disease. This relationship needs to be maintained across multiple distinct cohorts and be observable regardless of confounding factors (e.g., age, sex, ethnicity, diet, etc.) or comorbidities. The limited availability of both replicate studies and the associated meta-analysis presents a fundamental challenge to accurately assessing the reliability and reproducibility of potential molecular biomarkers. Further, the lack of a comprehensive evaluation of these clinical studies hinders the identification of study design parameters that may negatively impact the reproducibility of the outcomes. To address these issues, we present a detailed analysis of 24 clinical pancreatic ductal adenocarcinoma (PDAC) metabolomics studies [6,7,8,9,10,11,12,13,14,15,16,17,18,19,20,21,22,23,24,25,26,27,28,29]. Our meta-analysis revealed an extensive variation in these studies and highlighted a substantial inconsistency in outcomes, but also identified a subset of metabolites that were identified across multiple datasets. The analysis also exposed study design features that likely contributed to a potentially high false positive rate and exposed the need to establish best practices to successfully identify predictive molecular biomarkers.

Over 48,000 deaths from pancreatic cancer occur each year in the U.S, where PDAC accounts for more than 90% of all pancreatic cancers [30,31]. Pancreatic cancer has the lowest five-year survival rate among all cancers (11%) and is the third leading cause of cancer deaths in the U.S. [31]. This poor prognosis largely stems from the inability to identify the disease until the onset of late-stage symptoms when the cancer has metastasized, and surgery is no longer an option [32,33,34]. In the absence of surgery, there are limited alternative treatment options for pancreatic cancer (i.e., gemcitabine and folfirinox) [35], where a limited increase in survivability and drug resistance are common outcomes [36]. There is an urgent need to identify molecular biomarkers to assist in the diagnosis and prognosis of PDAC and to aid in monitoring a patient’s response to treatment. To date, there have been 24 PDAC metabolomics studies conducted to identify potential metabolite biomarkers [6,7,8,9,10,11,12,13,14,15,16,17,18,19,20,21,22,23,24,25,26,27,28,29]. 

Metabolomics is experiencing an exponential growth that includes extensive usage for biomarker discovery [37]. Metabolomics offers promising opportunities for developing new diagnostic and prognostic tools and consequently is of great interest to clinicians [38]. Nevertheless, there are still many challenges faced by the metabolomics community, which include successfully transitioning laboratory results to the clinic. This issue is clearly seen by the drastic variation in metabolite biomarkers identified between similarly diseased patients. These variations can be attributed to multiple factors. Sample collection and storage, cohort demographics, data processing, and analytical analyses can all impact a study’s outcome [39,40]. Due to the inherent complexity of patient samples, it is vital that these factors be controlled and minimized to achieve reliable and robust results. Simply, the natural variation among biological replicates can easily obfuscate any disease-dependent signal, which would negate the utility of the molecular biomarker. 

## 2. Materials and Methods 

### 2.1. Search Method

The scientific literature was searched using the Web of Science database for clinical PDAC metabolomics studies using the following queries: “pancreatic cancer and clinical metabolomics”, “pancreatic ductal adenocarcinoma and clinical metabolomics”, “pancreatic cancer and metabolomics”, and “pancreatic ductal adenocarcinoma and metabolomics”. The following manual selection criteria were then used to refine the list of metabolomics studies for the meta-analysis: (i) only human samples, (ii) a control group that was comprised of either healthy individuals, healthy pancreatic tissue, or chronic pancreatitis, and (iii) PDAC comparisons to other diseases including various cancers were omitted. Some studies were also omitted that fit the above criteria but were not accessible. 

### 2.2. Data Compilation

Experimental parameters and the metabolomics results from each study were compiled into an Excel file for comparison and statistical analysis. These parameters included: number of samples or patients, types of control samples, analytical technique, type of data collected, number of metabolites identified as significant after analyses, choice of *p*-value for statistical significance, fold change (FC) values, FC direction, data deposition, and inclusion of patient demographic data. Where appropriate, all values were tabulated by calculating sums, averages, standard deviations, and histogram distributions.

### 2.3. Statistical Analysis

All metabolites identified as significantly changing with their associated FC value and FC direction were combined into a single list in Excel. For further comparison of the results from the studies, 20% of the entire data set or a sub-grouping was used as an arbitrary cutoff to identify recurring metabolites and to create a list of metabolites that were significantly changing across multiple studies. FC values that were Log2 transformed in the original study were converted back to their original values. The FC values of metabolites that were identified in at least 20% or five papers were then averaged and a standard deviation calculated to demonstrate the variation in FC across studies. A median for all FC values was calculated based on analytical technique. 

## 3. Results and Discussion

A total of 24 metabolomics studies focused on identifying disease biomarkers from PDAC patients were compiled [6,7,8,9,10,11,12,13,14,15,16,17,18,19,20,21,22,23,24,25,26,27,28,29]. The techniques used in each study, source of sample, and number of significantly identified metabolites are detailed in Table 1. The demographic data of these studies is summarized in Table 2. Mass spectrometry (MS) was the predominant analytical platform for 17 studies; whereas nuclear magnetic resonance (NMR) was employed in 6 studies, and only one study used both NMR and MS. The data collected to identify metabolites also varied between each technique and study. MS studies differed in the use of MS and MS/MS data in which some studies only used MS data for identification, while other studies started with MS and then proceeded with tandem MS for a more targeted analysis. Alternatively, some MS studies used tandem MS for the entire data collection. NMR studies primarily employed one-dimensional (1D) ^1^H NMR spectral data, with only two studies supplementing the 1D ^1^H NMR data with two-dimensional (2D) spectral data. 

Qualitative metabolite changes (i.e., only increase or decrease) versus quantitative fold changes, health status of control samples, sample sources, and cohort size also varied greatly across these studies. All these confounding factors are likely to impact the metabolites that are detected and the statistical significance of any observable difference. For example, exosomes (1 study), pancreatic tissues (3 studies), plasma (4 studies), serum (13 studies), or urine samples (3 studies) were used in the 24 PDAC biomarker studies. Naturally, some metabolites may be sample-specific and the different sample types may yield divergent metabolic profiles. However, 71% of the studies did use serum or plasma as a biomarker source, greatly diminishing this concern. Since most of the studies used the same type of biological sample (i.e., plasma or serum derived from blood), a true disease-dependent biomarker may be expected to be identifiable across studies.

An observable change in a metabolite is completely dependent on the reference or control group. The 24 PDAC studies used different control groups corresponding to healthy controls (16 studies), chronic pancreatitis or benign pancreatic disease (5 studies), adjacent, healthy tissues from PDAC patients (2 studies), and patients with benign hepatobiliary disease (1 study). The power or statistical significance of a metabolomics study is also dependent on the cohort size, which ranged from 5 to 360 patients across the 24 studies [41,42]. How the spectral data was analyzed may also impact the reliability and reproducibility of the observed metabolite differences. Over 70% (17) of studies relied on a simple qualitative analysis of the data and only seven studies used a more comprehensive quantitative interpretation of the spectral data. Specifically, a qualitative analysis means the paper only reported if the metabolite was increased or decreased in pancreatic cancer patients. Conversely, a quantitative analysis reported both the magnitude and direction of the fold change in the metabolite relative to controls.

It is important to note that several studies had incomplete or absent patient demographic data. Data deposition was also largely lacking across these studies despite the fact that data deposition is highly valued by the metabolomics community. This is evidenced by the recent growth in data repositories (e.g., The Metabolomics Workbench, https://www.metabolomicsworkbench.org/ (accessed on 8 August 2022)), which enable the routine submission of metabolomics datasets. Four of the eight studies published within the last two years did list data deposition information. While data deposition is improving and becoming common, it is not universal and would benefit by publishers requiring the deposition of metabolomics data.

A total of 655 unique metabolites (Appendix A) were identified as significantly changing across the 24 PDAC biomarker studies. Of these 655 metabolites, 40 were detected in urine, 68 in plasma, 339 in serum, and 279 in the remaining sample types (Appendix A). Of course, some metabolites were detected in more than one type of biofluid. A heatmap summarizing the distribution of these 655 metabolites across the 24 PDAC studies is shown in Figure 1. It is readily apparent from this heatmap that the metabolites are randomly dispersed. The hierarchical clustering based on the PDAC studies failed to yield any discernable groupings. Similarly, studies using the same sample type or analytical technique did not demonstrate any clustering. Clearly, the choice of sample type or analytical technique was insufficient to explain the large metabolite variability. This suggests that other factors, including study design and experimental parameters, may have an equal to or greater than impact on the reproducibility and accuracy of the metabolomics results.

Of the 655 unique metabolites, 87% of these metabolites were only identified in a single study (Figure 2A). As the occurrence rate (number of studies detecting the same metabolite) increased, the number of recurring metabolites across multiple studies rapidly decreased. The highest recurring metabolites were present in less than half of the studies with less than 1% of the metabolites detected in seven or more studies. These trends hold true even when the data is sub-grouped by sample type (Appendix A). Thus, an arbitrary cutoff of 20%, in which the same metabolites were identified in five or more papers, was used to compile a list of recurring or reproducibly identified metabolites (Table 3). Only 16 metabolites (2.4%) were identified by at least five papers as significantly changing between healthy controls and PDAC patients. Notably, glutamate (11 studies), glutamine (10 studies), and ornithine (9 studies) were the metabolites identified in most studies and all have established roles in cancer [43,44,45].

The average FC values and the direction of the fold change were also tabulated (Table 3, Appendix A). Notably, a majority of the 655 metabolites were only identified by MS with a mere 7% (46) of the metabolites only identified by NMR and 35 (5.3%) metabolites identified by both platforms (Figure 2B). This disparity is not surprising given the limited dynamic range and sensitivity of NMR compared to MS [46]. The median magnitude change in FC values (i.e., ignoring direction) for all metabolites detected by MS or NMR was 1.85 and 1.16, respectively. A histogram of FC values is plotted in Figure 3A, where numbers greater than one indicate a relative increase in PDAC and numbers less than one indicate a decrease in PDAC. Most FC values were less than two. This is concerning, especially for MS data, where an FC > 2 or 3 is a commonly used criterion of likely significance. The routine use of a lower FC value to define a threshold for a statistically significant metabolite change may partly explain the low recurrency of metabolites across the 24 PDAC studies. 

The cohort size varied greatly across the PDAC biomarker studies with participant numbers ranging from fewer than 10 patients to over 100 (Figure 3B). A majority of the metabolomics studies only disclosed qualitative metabolite changes (Figure 3C) while the choice of *p*-value that indicated the statistical significance of the FC varied greatly from *p* < 1 × 10^−5^ to *p* < 0.3 (Figure 3D). It was also concerning that only 8 of the 24 studies reported the use of false discovery rate (FDR) corrected *p*-values or noted the type of FDR method that was applied. The direction of metabolite FCs across the 24 PDAC biomarker studies was also inconsistent. Only 69% of metabolites identified by at least two studies exhibited the same FC direction (Figure 3E). Metabolites that were identified in at least three studies showed only a 53% consistency in FC direction (Figure 3F), which continued to drop to a consistency of 38% for metabolites identified in at least five studies (Figure 3G, Table 3).

The number of statistically significant metabolite changes were plotted as a function of cohort size and *p*-value in Figure 4. Overall, there was not a clearly identifiable relationship between these three parameters. As cohort size increased, the number of significantly changing metabolites did decrease, but there was a large random variation at smaller cohort sizes (<100). The larger cohort sizes initially suggested a more robust evaluation of metabolites that were changing between the disease and control groups since the number of identified metabolites stabilized, but the *p*-values for the larger cohort studies also varied greatly from *p* ≤ 0.05 to *p* < 0.3. This raised legitimate concerns about the reliability of the apparent consistency in the number of identified metabolites in the larger cohort studies. Some metabolites may be included or removed from the list of retained metabolites by the simple choice of *p*-value, which cannot be easily rectified. Thus, the overall variability in *p*-values presents an interesting challenge in comparing the reproducibility of metabolites across multiple clinical studies and raises some serious questions. How do we achieve statistical confidence and normalize across multiple clinical datasets without negating the underlying value of the clinical research? One approach to address this issue is for the community to adopt a single standard of statistical significance that must be adhered to by published clinical studies. The 24 PDAC biomarker studies hint at a potential consensus since 16, or 67%, of the studies, utilized a *p*-value of ≤0.05. The remaining eight studies were equally divided between using either a higher or lower *p*-value. But is a *p*-value of ≤ 0.05 still too aggressive?

A proper balance needs to be obtained that accommodates the complexities, difficulties, and potentially large biological variabilities encountered within human samples to avoid false negatives or Type II errors (argument for larger *p*-values and smaller FC values) while simultaneously avoiding an abundance of false positives or Type I errors (argument for smaller *p*-values and larger FC values) that lead to erroneous disease biomarkers. Neither an abundance of false positives nor false negatives is a desirable outcome, but in the context of identifying a reliable and reproducible biomarker to diagnose and monitor a disease against a complex human background, it would be preferable to initially minimize false positives. Since 87% of the metabolites identified from the 24 PDAC studies were only observed by one study and are likely false positives, it appears the clinical metabolomics field is currently heavily biased toward Type I errors. In other words, the community may be better served by adopting a more restrictive *p*-value and FC value. Of course, the choice of *p*-value or FC value alone doesn’t account for the apparently high false positive rate as other experimental factors are likely contributors.

Sample storage and handling, data collection and analysis, and cohort demographics are some of the experimental design factors that may influence the successful outcome of a biomarker discovery project. For example, time of sample pre-processing, length of storage, sample preparation protocols, patient demographics, and bioinformatic analyses are important factors in metabolomics studies that have all been previously shown to impact study outcomes [47,48,49,50,51]. The metabolome is chemically unstable and will change as a function of time and storage conditions. Most of the PDAC biomarker studies cited herein described storage conditions and sample pre-processing protocols but did not define the length of time that passed before samples were prepared for analyses and spectral data were collected. Cohort demographics can also influence metabolic profiles with differences in age, gender, and pre-existing health conditions, among other factors, inducing large biological variances that negatively impact biomarker identification. Simply, the within-group difference (i.e., biological variability) is greater than the between-group difference (i.e., difference between healthy controls and PDAC patients). These problems are compounded when comparing results across multiple clinical studies, which further emphasizes the need to establish best practices and to standardize protocols for metabolomics clinical studies.

To account for these and other confounding factors, three different sub-groupings of the entire dataset of 24 PDAC studies were made. These sub-groupings included: (A) 13 serum-only studies (Appendix A), (B) 4 studies with the largest cohort size (>90 participants), and (C) 20 studies with a *p*-value ≤ 0.05. A recurring metabolite was identified by being in at least 20% of the studies in a sub-group or in a minimum of two studies, whichever was greater. Using these criteria, the full data set and sub-groups A and C had 16 recurring metabolites each, while sub-group C had 15 metabolites. Overall, 23 unique metabolites were identified by combining the three sub-groups with the full data set (Table 4). In total, seven new metabolites were identified by the three sub-groups that were not previously identified using the full data set. In all but one case, the metabolite was identified by only one sub-group. Conversely, and potentially more importantly, 10 of the recurring metabolites identified with the full data set were also identified in the three sub-groups. This list of 10 metabolites (i.e., alanine, arginine, creatine, glutamate, glutamine, histidine, lysine, ornithine, phenylalanine, and threonine) may be the most robust and reliable choice for potential PDAC biomarkers from the 24 metabolomics studies. The six metabolites that were only identified by one of the three sub-groupings or the full data set are likely false positives. Further, the fact that the three sub-groups reduced the number of likely biomarkers from 16 to 10 strongly infers that cohort size, choice of *p*-value, and consistency of sample source all play a role in the proper identification of reproducible metabolite biomarkers. Despite all these challenges and as summarized in Table 3 and Table 4, there was still a subset of metabolites that were detected across multiple clinical metabolomics studies that have the potential to serve as biomarkers for PDAC.

While the consistency of identified metabolites was low, the metabolites identified in at least 20% of the PDAC studies all have important roles in carcinogenic onset and progression. For example, the most identified metabolite, glutamate, has been shown to promote pancreatic cancer cell invasion and migration [52]. Similarly, glutamine is used by cancerous cells as an alternative fuel source for TCA via glutaminolysis and is vital to many types of cancers, including pancreatic cancer [53]. Glucose is a classic hallmark of oncogenesis with increased consumption in cancer cells promoting cellular proliferation and survival, commonly called the Warburg Effect [54]. Ornithine is a proposed biomarker of cancer as increased levels of ornithine metabolism have been evident in studies of tumor-bearing animals [55]. Creatine has been shown to promote cancer metastasis and to decrease animal survival in colorectal and breast cancer [56]. Citrate is an important intermediate in the TCA cycle and is rapidly consumed by cancerous cells to promote cellular proliferation and drug resistance [57]. The remaining 10 amino acids that were commonly identified by these PDAC studies and include alanine, arginine, asparagine, aspartate, histidine, lysine, phenylalanine, proline, threonine, and tyrosine, serve as alternative energy sources and contribute to redox balance [58]. Again, the fact that this set of consistently identified metabolites was also associated with cellular processes relevant to cancer increases the likelihood of providing a reliable diagnosis of PDAC. Nevertheless, some of these metabolites did not change consistently across the clinical studies, which is problematic. Simply, for a metabolite to be a true biomarker for PDAC it should be consistently detected in a biofluid and repeatedly demonstrate the same relative change in abundance. Overall, clinical metabolomics has made important progress in the identification of potential metabolic biomarkers for PDAC, but there are still aspects of the process that must be addressed to increase the robustness and the reliability of these outcomes.

## 4. Conclusions

This meta-analysis has highlighted the large variation in potential metabolite biomarkers observed across 24 PDAC metabolomics studies. A surprisingly high number of 655 metabolites were identified where 87% of these metabolites were only observed by one study, suggesting an extremely high false positive rate. There was no discernable difference in this variability based on sample type or analytical method. Only 16 metabolites were identified by 20% or more of these studies. The most consistently identified metabolite (glutamate/glutamic acid) was only listed in 11 of the 24 studies. Only 6 of the 16 metabolites had the direction (increase or decrease) of their fold change consistently reported. When cohort size, sample type, and *p*-value were considered, the number of potentially reproducible metabolites dropped from 16 to 10. Differences in cohort size and demographics, *p*-value choice, fold-change significance, sample type, handling and storage, data collection, and analysis were all factors that likely contributed to the large inconsistencies between these 24 PDAC metabolomics studies. Taken together, a strong case is made that a high level of consistency in experimental design is necessary to accurately identify reproducible disease biomarkers across multiple studies. In short, a conservative approach is demanded, such as a large cohort size (>100), small FDR-corrected *p*-values (<0.01), and large FC values (>2–3) that also necessitates the establishment and adherence to best practices for metabolomics clinical studies. Otherwise, the scientific community will be awash in irreproducible, erroneous, and contradictory results that will likely undermine the utility and beneficial impact of metabolomics in a variety of fields including disease diagnosis and progression. While this meta-analysis herein focused on clinical studies of PDAC patients, the shortcomings highlighted are not exclusive to pancreatic cancer and are likely to be observed in numerous other clinical disease studies.

Despite the highlighted challenges and problems, it is important to note that a comprehensive analysis of these 24 PDAC studies did identify a list of 10 potential metabolite biomarkers that obtained a level of reproducibility across multiple studies. The fact that these metabolites were identified across these multiple studies with divergent experimental parameters and conditions provides a higher level of statistical confidence than can be achieved by a single study. The availability of multiple studies performed by different researchers using samples from a different cohort of patients is invaluable to assess both the true reproducibility of potential disease biomarkers and provide a path forward to improve our approach to discovering biomarkers through clinical metabolomics.

## Figures and Tables

**Figure 1 cancers-14-03992-f001:**
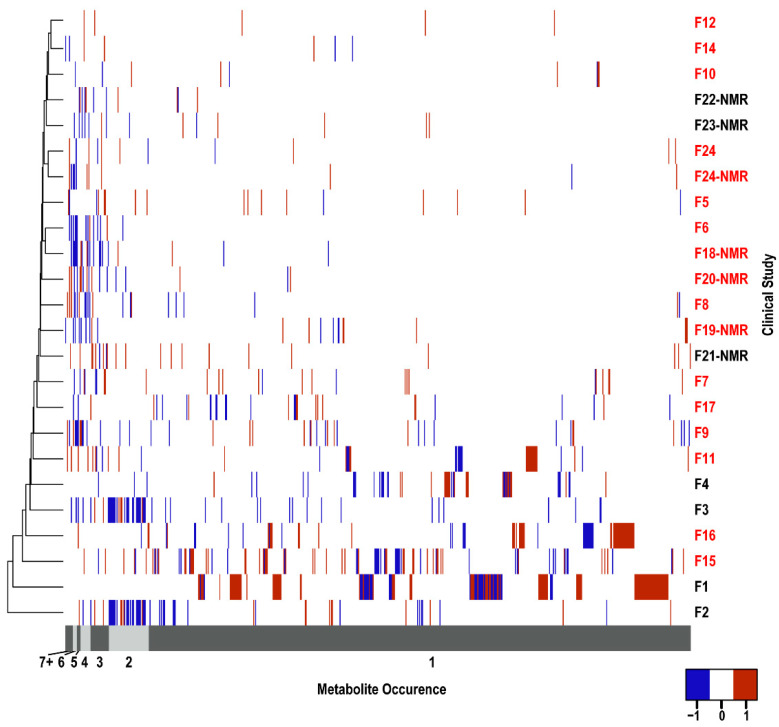
Heatmap of Metabolite Occurrence. A heatmap depicting the distribution of the 655 metabolites identified by the 24 PDAC papers. A hierarchical cluster of the clinical studies was performed where the row numbers correspond to the papers listed in Table 1. The numbers colored red corresponds to studies that used either plasma or serum samples. The NMR label identifies studies that used NMR whereas studies that used mass spectrometry are unlabeled. Metabolite occurrence order corresponds to the groupings in Figure 2A. The exact list and order of the metabolite names can be found in Appendix A. For all metabolites with only a single occurrence, the metabolites are ordered alphabetically. Since most studies did not report a quantitative fold change, only a qualitative increase (red, +1) or decrease (blue, −1) in the metabolite relative to a control is indicated. Metabolites not identified in a study were assigned zero and are white.

**Figure 2 cancers-14-03992-f002:**
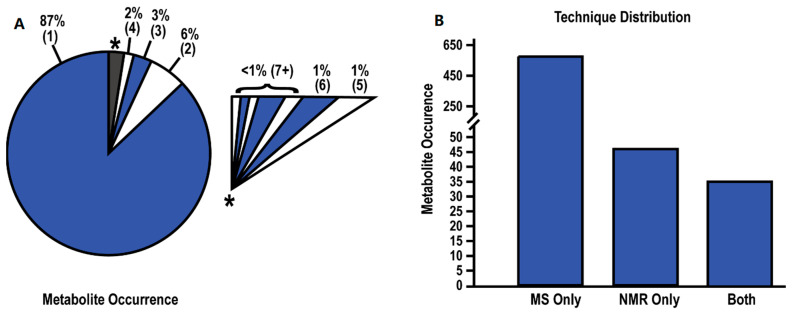
Summary of Metabolite Data. (**A**) Pie chart depicting the rate of metabolite occurrence across the 24 studies. Numbers within parentheses indicate the number of studies identifying the metabolites. An expanded view of the grey-starred slice of the pie chart is shown as an insert. (**B**) Bar chart depicting the number of metabolites identified by MS, NMR, or both techniques.

**Figure 3 cancers-14-03992-f003:**
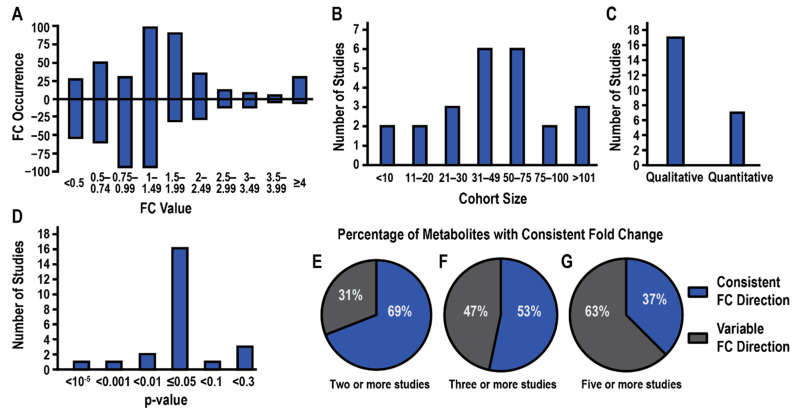
Summary of Experimental Parameters, Design, and Outcomes. (**A**). Bar chart showing the distribution of FC values across the 24 studies. A value greater than one indicates a relative increase in the metabolite in PDAC while a value less than one indicates a relative decrease in the metabolite in PDAC. (**B**). Bar chart showing the distribution of cohort size across the 24 studies. (**C**). Bar chart depicting the number of studies that reported a quantitative or qualitative metabolite fold change. (**D**). Bar chart showing the distribution of *p*-value used to define a statistically significant metabolite change. One study did not report a *p*-value. Pie charts summarizing the consistency in reported fold change direction (increase or decrease) for metabolites identified in (**E**). two or more studies, (**F**). three or more studies, or (**G**). five or more studies. Blue indicates percentage of metabolites with the same fold change direction in all studies and gray indicates percentage of metabolites with variable fold change directions.

**Figure 4 cancers-14-03992-f004:**
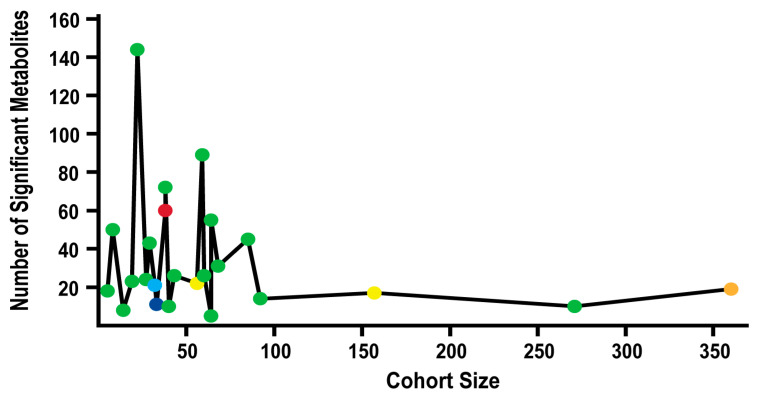
Significant Metabolites, Cohort Size, and *p*-value. Line chart comparing cohort size and the number of metabolites identified as significantly changing. Circle colors correspond to *p*-value used in the study to identify statistical significance: *p* < 1 × 10^−5^ (dark blue), *p* < 0.001 (light blue), *p* < 0.01 (red), *p* ≤ 0.05 (green), *p* < 0.1 (orange), and *p* < 0.3 (yellow). One study was excluded since it did not report a *p*-value to determine statistical significance.

**Table 1 cancers-14-03992-t001:** Paper Summaries.

Technique	Paper	Fluid	Metabolites ^a^
MS	1 ^Tao et al.^	Exosomes	144
	2 ^Budhu et al.^	Pancreatic tissue	60
	3 ^Zhang et al.^	Pancreatic tissue	55
	4 ^Unger et al.^	Pancreatic tissue	50
	5 ^Urayama et al.^	Plasma	18
	6 ^Fukutake et al.^	Plasma	19
	7 ^Luo et al.^	Plasma	26
	8 ^Itoi et al.^	Serum	24
	9 ^Kobayashi et al.^	Serum	45
	10 ^Mayerle et al.^	Serum	10
	11 ^Iwano et al.^	Serum	43
	12 ^Lindahl et al.^	Serum	5
	13 ^Di Gangi et al.^	Serum	10
	14 ^Xiong et al.^	Serum	8
	15 ^Martín-Blázqeuz et al.^	Serum	89
	16 ^Macias et al.^	Serum	72
	17 ^Wang et al.^	Serum	31
NMR	18 ^Michálková et al.^	Plasma	26
	19 ^Zhang et al.^	Serum	23
	20 ^Bathe et al.^	Serum	22
	21 ^Davis et al.^	Urine	21
	22 ^Napoli et al.^	Urine	11
	23 ^Sahni et al.^	Urine	14
Both	24 ^McConnell et al.^	Serum (MS)	17
	24 ^McConnell et al.^	Serum (NMR)	16

^a^ Total metabolites identified as significantly changing after statistical analyses.

**Table 2 cancers-14-03992-t002:** Summary of Study Designs from the 24 PDAC Papers.

Analytical Techniques				
**NMR**	**MS**		**Both**	
6	17		1	

**NMR Data**				
**1D**	**1D & 2D**			
5	2			

**MS Data ^a^**				
**MS**	**MS/MS**		**MS and MS/MS**	
4	8		5	

**Fold Change**				
**Qualitative**	**Quantitative**			
17	7			

**Data Deposited ^b^**				
**Yes**	**No**			
5	19			

**Control Samples**				
**Healthy/normal** **control**	**Chronic** **pancreatitis/benign** **pancreatic disease**		**Adjacent, non-tumor** **tissue**	**Benign hepatobiliary disease**
16	5		2	1

**Sample Sources**				
**Urine**	**Serum**	**Plasma**	**Tissue**	**Exosomes**
3	13	4	3	1

^a^ Fragments used to identify metabolites; ^b^ Includes data that is available upon request to authors.

**Table 3 cancers-14-03992-t003:** Most Commonly Occurring Metabolites ^a^.

Metabolite	# of Publications	Higher Abundance Relative to Control	Lower Abundance Relative to Control	Average FC Relative to Control	STD	STD/Avg	Outlier Values	# of Actual Values (Including Outliers)
Glutamate/glutamic acid	11	9	2	(2.37) 0.61	(0.83) 0.06	(0.35) 0.09	NA	(5) 2
Glutamine	10	0	10	0.70	0.13	0.19	NA	7
Ornithine	9	3	6	(1.43) 0.51	(0.34) 0.02	(0.24) 0.03	NA	(3) 3
Lysine	8	1	7	(1.03) 0.67	(0) 0.15	(0) 0.22	NA	(1) 5
Phenylalanine	8	4	4	(1.17) 0.74	(0.09) 0.17	(0.07) 0.23	NA	(2) 4
Threonine	8	1	7	0.76	0.15	0.20	NA	4
Arginine	7	3	4	(1.26) 0.61	(0.30) 0.05	(0.24) 0.09	NA	(3) 2
Proline	7	0	7	0.74	0.15	0.20	NA	4
Alanine	6	1	5	(1.23) 0.62	(0) 0.06	(0) 0.10	NA	(1) 2
Creatine	6	0	6	0.53	0.27	0.51	NA	4
Histidine	6	0	6	0.67	0.08	0.12	NA	5
Tyrosine	6	0	6	0.68	0.14	0.21	NA	3
Asparagine	5	0	5	0.79	0.13	0.17	NA	4
Aspartic acid/aspartate	5	2	3	(2.47) 0.82	(0.31) 0.16	(0.13) 0.20	NA	(2) 2
Citrate	5	2	3	(2.10) 0.44	(1.03) 0.13	(0.49) 0.30	NA	(2) 2
Glucose	5	4	1	(1.28) 0.22	(0.23) 0	(0.18) 0	(421.58)	(3) 1

^a^Values in parentheses are associated with increased fold changes. Values outside parentheses are associated with decreased fold changes.

**Table 4 cancers-14-03992-t004:** Common metabolites within study sub-groupings.

Metabolite	Full 24	Serum Only	Largest Cohorts	*p* ≤ 0.05
3-hydroxybutyrate		X		
Alanine	X	X	X	X
Arginine	X	X	X	X
Asparagine	X	X		
Aspartic acid/aspartate	X	X		X
Citrate	X			X
Creatine	X	X	X	X
Creatinine				X
Glucose	X			X
Glutamate/glutamic acid	X	X	X	X
Glutamine	X	X	X	X
Glycocholic acid		X		X
Hippurate/hippuric acid				X
Histidine	X	X	X	X
Isoleucine			X	
Lysine	X	X	X	X
Myoinositol			X	
Ornithine	X	X	X	X
Phenylalanine	X	X	X	X
Proline	X	X	X	
Threonine	X	X	X	X
Tyrosine	X	X	X	
Urea			X	
**Sum**	**16**	**16**	**15**	**16**

## Data Availability

A compilation of the 24 PDAC papers and the Excel files summarizing the experimental parameters and metabolomics results are available from the authors.

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
