# Peer review of "Meta-Analysis Reveals Both the Promises and the Challenges of Clinical Metabolomics"

_cancers, 2022, doi:10.3390/cancers14163992_

Round 1

Reviewer 1 Report

The authors have addressed most of the comments and, this study will help to get the information to the other researchers in the same field of identifying biomarkers in PDAC. However, the authors did not address a part of the previous comment mentioned by reviewer 1, which is Comment 1.: "Further, they can consider dividing and representing the metabolites in plasma, serum, and tissue."

The information of biomarkers derived from different sample types (urine, serum/plasma, tissue, and exosomes) is also valuable information. If the data representation in the manuscript is difficult due to the disperse of the data across studies, authors could have mentioned the commonly identified biomarkers among sample types in the manuscript as a text and/or at least mention the sample type under the study/paper number in the supplementary table. This information is really helpful for future studies and researchers in the same field.

So, please mention a few sentences about the biomarkers identified across different sample types (either common ones or if not, please mention) and, please put the sample type (urine, serum/plasma, tissue, and exosomes) in the supplementary table under the paper number.

Author Response

Reviewer 1

  1. However, the authors did not address a part of the previous comment mentioned by reviewer 1, which is Comment 1.: "Further, they can consider dividing and representing the metabolites in plasma, serum, and tissue." So, please mention a few sentences about the biomarkers identified across different sample types (either common ones or if not, please mention) and, please put the sample type (urine, serum/plasma, tissue, and exosomes) in the supplementary table under the paper number.

Response: The Supplemental Table 1 has been updated according to the Reviewer’s request. Specifically, the sample type, first author name, and analytical method has been added as descriptors for each of the 24 papers summarized in the Excel file. Further, summation data has been added to the worksheet, which counts the number of detected metabolites per column, and the number metabolite matches per row (across papers). Also, four additional worksheets have been added to the Excel file, where each new worksheet summarizes the metabolites detected in plasma, serum, urine , and others. Metabolites must be detected by at least on paper to be listed in the subgroup tables. Also, the metabolites are ordered (from high to low) by the number of matches in the subgroup tables. There are now supplemental Tables 1a-e.

The manuscript has been modified throughout to cite the new supplemental tables, to identify the number of metabolites detected per sample type, and to acknowledge that there is no discernable difference in the data variability based on type of sample.

Reviewer 2 Report

Re: Results vs Discussion sections: Technically, Results section should state the findings of the research without interpretations. Discussion section should include interpretations based on the findings. If considered for combining the two sections, the subtitle should reflect that. 

Author Response

Reviewer 2

  1. Re: Results vs Discussion sections: Technically, Results section should state the findings of the research without interpretations. Discussion section should include interpretations based on the findings. If considered for combining the two sections, the subtitle should reflect that.

Response: The subtitle has been corrected to indicate both Results & Discussion.

Reviewer 3 Report

This manuscript by Roth and Powers analysing the consistency of potential PDAC metabolomic biomarkers is well written and raises important concerns regarding the repeatability of experimental metabolomic results across studies. This is my second time reviewing this manuscript, as I reviewed the first submission of this manuscript, and I believe the authors have adequately addressed any suggestions made by reviewers after review of the first submission. I recommend this manuscript for acceptance with the minor revision below.

Figure 3: I would suggest a label for panel E, F and G. Such as “Percentage of metabolites with similar fold-change direction” and then a specific label for E,F and G indicating how many studies were considered. (e.g. two or more, three or more, etc). I would also include a legend to the right of panel G to indicate the meaning of gray and blue areas.

Author Response

Reviewer 3

  1. Figure 3: I would suggest a label for panel E, F and G. Such as “Percentage of metabolites with similar fold-change direction” and then a specific label for E,F and G indicating how many studies were considered. (e.g. two or more, three or more, etc). I would also include a legend to the right of panel G to indicate the meaning of gray and blue areas.

Response: Figure 3 has been updated as suggested.

This manuscript is a resubmission of an earlier submission. The following is a list of the peer review reports and author responses from that submission.

Round 1

Reviewer 1 Report

The manuscript by Roth et al. shows a meta-analysis of clinical metabolomics related to PDAC. The authors identified ten metabolites that may serve as biomarkers. The study is largely straightforward, but the below-mentioned points could be addressed to represent the scope of the study.

Major:

1.  Data represented in the manuscript is mainly for experimental parameters (Ex. Figure 2) and the majority of the figures do not represent the information of metabolites. This may distract the scope of the study, which is to represent the possible biomarkers and their expression. Only Tables 3 and 4 give the information about the metabolites. Authors may use heat maps, and volcano plots to represent data. Further, they can consider dividing and representing the metabolites in plasma, serum, and tissue. In summary, the authors could have given more attention to representing the metabolites in PDAC in a proper way in the manuscript.

2. The explanation in the results section is mainly focused on geographic data, cohort size and etc. and the information related to metabolites and PDAC is lacking. Especially, the association between identified metabolites and PDAC is not clearly illustrated in the results/discussion section.

3. The challenges in the meta-analysis are well discussed, but the promises are not illustrated well.

4. Authors may consider performing class prediction analysis and validation in independent data sets to discover candidate metabolites using these data sets.

Minor:

Method section can be divided into sub-sections like search method, data/statistical analysis and etc.

Reviewer 2 Report

General: This manuscript presents the meta-analytic review of metabolome based pancreatic cancer detection biomarker studies. Literature search and identification of 24 metabolomics pancreatic cancer biomarker studies were described and various perspectives are discussed with pitfalls and potential improvements to be sought for future studies. This is a useful summary of the current status of the field. Due to the nature of variabilities in methodologies including the samples, experimental protocol, statistical analyses, the difficulty in direct comparison of studies and making a meaningful conclusion is encountered. Consequently, the article illustrates the inconsistencies of metabolomics studies in pancreatic cancer biomarker research.

Issues: 

1.     Of the various potential variables, which are mentioned, with contribution to the inconsistencies in the metabolomics biomarker discovery studies, it would be useful to provide a set of strongly weighted, important ones to include – based on past studies (such as dealing with patient population (including consistent controls), sample storage, type of samples, statistical consideration, etc.) - in standardization of future studies, which would lead to more comparable studies from which to perform more adequate meta-analysis. 

2.     Separate Discussion section from Results desired.

Reviewer 3 Report

Peer Review of Meta-analysis Reveals Both the Promises and the Challenges of Clinical Metabolomics

General comments:

Roth and Powers have written a generally clear and concise meta-analysis regarding the important question of metabolite biomarker validation for the diagnosis and characterization of pancreatic ductal adenocarcinoma. The authors explore the important topic of reproducibility in metabolomics that will need to be addressed by the field in the future, especially if it is to be used as a tool for clinical diagnosis. For this reason I recommend this paper for publication if the following minor revisions are addressed:

Abstract comments:

Line 24-27: Line starting “Unfortunately, a large variability was also observed…”. I believe the authors are trying to say that the 24 studies in sum identified 655 metabolites that are potential biomarkers with most (87%) being identified by single studies (albeit not necessarily all in the same individual study). The start of this sentence is confusing. Perhaps the authors could write something similar to: “However, 87% of the 655 metabolites identified as potential biomarkers were identified in single studies.” I think this makes the potential lack of reproducibility/variability in the identification of these biomarkers implicit.  

Line 29: Is it a potential false positive rate? It could be that some of the single-study identified metabolites are true positives, albeit requiring certain experimental conditions to detect consistently.

Introduction comments:

Line 42: “and chemicals that also includes metabolites”. Perhaps molecules could be a better word here instead of chemicals?

Results comments:

Line 102: Would the authors be opposed to using MS and MS/MS instead of MS1 and MS2 to indicate tandem mass spectrometry? Due to the journal’s referencing style it looks like a reference rather than an abbreviation for tandem mass spec.

Line 124: When the authors write “sample sources of plasma and serum”. I believe the authors mean the use of consistent sample types rather than the sources from which the plasma and serum were obtained. However, I may be mistaken. I suggest rewriting the sentence to clear up this ambiguity in phrasing.

Line 126: I suggest removing the word “also” in “…metabolite is also completely dependent…”

Line 131: “The power or statistical significance of a metabolomics study is also dependent on the cohort size, which ranged from 5 to 360 patients across the 24 studies.” This sentence could benefit from a reference to a statistical paper discussing this concept.

Line 135-137: Please expand on what is meant by a simple qualitative analysis versus a more comprehensive quantitative interpretation.

Line 155: I would suggest rephrasing this as “Thus, an arbitrary cut off of 20%, in which the same metabolites were identified in 5 or more papers, was used to compile a list of recurring or reproducibly identified metabolites.”

Line 160: I suggest removing the word reassuringly.

Line 183: I am unclear by what the authors mean by “The routine use of lower FC values” do they mean the routine reporting of lower FC values as significant in the 24 studies?

Line 179: I understand what Figure 2A is trying to show (i.e. The distribution of fold-changes measured for the different metabolites in each study, with most occurring at FC values less than 2). However, this graph is difficult to interpret and the body text starting at this line does not make the interpretation of this figure much clearer. Are both axes meant to be labelled as p-value? Based on the body text description I would expect the x-axis to be labelled as FC and the y-axis to be labelled as FC occurrence.

Line 187: I am unclear by what the authors mean by qualitative and quantitative FC values. I assume the authors means something akin to relative quantities/concentration and absolute quantities/concentration of a metabolite being reported. A definition earlier in the text would be useful.

Line 188: In regards to figure 2D, could the authors expand on why some studies reported a p-value of 0.3 as significant? Is it possible that the p-values of 0.1 and 0.3 are FDR cut-off values of significance (e.g. a FDR cut-off value (q-value) of 0.1 after a Bejamini-Hochberg procedure is not unusual, however reporting a student’s t-test p-value of 0.1 might be unusual). Perhaps the authors could write about which statistical tests were used in each study and how this influenced the selection of a significance value cut-off.

Line 191: Should Only 69% of metabolites identified by two studies exhibited…” be “Only 69% of metabolites identified by at least two studies exhibited…”?

Table 1 comments:

Perhaps including the first author name et al is a better way to distinguish between the different papers in the paper column rather than assigning the reference another number? E.g. Paper 1 could be listed as Tao et al6.

Table 2 comments:

The full page spacing of this table could be distracting to the reader. I suggest making the columns more compact.

The table title of “Summary of Demographic Data” is confusing as the authors discuss patient demographic data in the results section. I suggest changing this table title to “Summary of study characteristics” or “Summary of study procedures” or another more apt title.

Table 3 comments:

Ensure that metabolite names fit on one line. For example phenylalanine has an e on the next line and glutamic acid is split in two lines after the gl.

Would another word instead of “Increasing” or “Decreasing” be more clear? Perhaps “higher abundance” and “lower abundance” or “higher amount” and “lower amount”? I would also suggest writing “Average FC relative to control”.

Figure 1 comments: Perhaps the authors could indicate where their blown up region beside the pie is located on the main pie.

Supplementary Table 1 comments:

It may be useful to colour code the metabolite names that are only detected in a single study for fast reading of the supplementary table.